# Spectral Riemann Surface Topology of Gapped Non-Hermitian Systems

**Anton Montag**[1,2]⋆, **Alexander Felski**[1]† **and Flore K. Kunst**[1,2]‡

**1** Max Planck Institute for the Science of Light, Erlangen, Germany
**2** Department of Physics, Friedrich-Alexander-Universität Erlangen-Nürnberg, Erlangen, German

⋆ anton.montag@mpl.mpg.de , † alexander.felski@mpl.mpg.de , ‡ flore.kunst@mpl.mpg.de

## Abstract

We show topological configurations of the complex-valued spectra in gapped non-Hermitian systems. These arise when the distinctive exceptional points in the energy Riemann surfaces of such models are annihilated after threading them across the boundary of the Brillouin zone. This results in a non-trivially closed branch cut that is protected by an energy gap in the spectrum. Their presence or absence establishes topologically distinct configurations for fully non-degenerate systems and tuning between them requires a closing of the gap, forming exceptional point degeneracies. We provide an outlook toward experimental realizations in metasurfaces and single-photon interferometry.

## 1 Introduction

Non-Hermitian systems feature properties with no counterpart in Hermitian models, such as skin states, dissipative phase transitions, and unidirectional transmission [1–3]. These features are closely tied to the topology of the complex-valued spectral structure of such systems [4, 5]. The extensive studies of non-Hermitian spectra have focused by-and-large on so-called exceptional points (EPs) [2, 6–16] and spectral winding numbers [17–20]. Here, we instead demonstrate topologically distinct configurations of the energy Riemann surfaces of fully non-degenerate non-Hermitian systems. For a two-dimensional periodic two-band system we show that four topologically distinct configurations are realized by closed non-contractible branch cuts in the Riemann sheet structure over the Brillouin zone. These configurations are structurally analogous to the ground state of the toric code, in which they represent a protected logical qubit pair within a toroidal spin lattice [21]. The $\mathbb{Z}_2 \times \mathbb{Z}_2$ topological order that protects the toric code ground states translates to a $\mathbb{Z}_2 \times \mathbb{Z}_2$ topological invariant for the Riemann sheet structure of the non-Hermitian lattice. We extend this analogy by showing that EPs in the non-Hermitian spectrum emerge similarly to excitations of the toric code [21–23]. While there are three types of excitations in the conventional toric code, the number of different excitations in the non-Hermitian model is tied to its band structure. We conclude with an outlook on potential experimental implementations in optical, plasmonic, and mechanical metasurfaces, and single-photon interferometers.

### 1.1 Non-Hermitian Bloch Hamiltonians and spectral Riemann surfaces

Two-dimensional lattice structures can be described by a Bloch Hamiltonian, which is defined over the toroidal surface formed by the two-dimensional Brillouin zone. Allowing for dissipative processes on the lattice results in a non-Hermitian Bloch Hamiltonian, $H(\boldsymbol{k}) \neq H^\dagger(\boldsymbol{k})$, and complex-valued eigenenergies. Topology in conventional Hermitian systems operates on the level of eigenstates, whereas in non-Hermitian systems topological structures may also emerge in the Riemann surfaces describing the energy spectrum [4, 18, 24]. The most prominent feature in non-Hermitian topology are exceptional points of order $n$ (EP$n$s), which cannot arise in Hermitian systems [1–4, 6]. At these points the spectrum of the Hamiltonian is $n$-fold degenerate and the EPs manifest as branch points in the energy surface structure. Consider a two-band system, described by a $2 \times 2$ Bloch Hamiltonian, which gives rise to two complex-valued energy sheets over the Brillouin zone. In this two-dimensional space, EP2s generically appear in pairs, see Fig. 1(a), with opposite spectral winding around the respective EP2 defining a topological charge [19, 20]. Any such pair is connected by a Fermi arc, an imaginary Fermi arc, and a branch cut of the Riemann sheets [4, 25–27]: (imaginary-)Fermi arcs are lines on which the real (imaginary) part of the eigenenergies is degenerate. Note that such arcs can also exist independent of EPs, for example at band touching lines, see Fig. 1(c). A branch cut, on the other hand, is a line along which a continuous multi-sheeted Riemann surface is separated into well-defined single-valued sheets; the path of such branch cuts can be chosen freely in general. Here we choose the common identification of the branch cuts with the Fermi arc that always connects the EP2 pair, and refer to the combined object as a *Fermi cut*. The presence of this Fermi cut is protected and results in a continuously connected energy structure. The protection originates from the presence of the stable EP2 pair and prevents the Fermi cut from being removed by any small perturbation. While such stable features of non-Hermitian systems have been studied, the presence of EPs implies spectral degeneracies that prevent the definition of a conventional topological gap. We here present a classification, resembling the topological structure of the toric code ground states, for which a spectral gap can be clearly defined nonetheless.

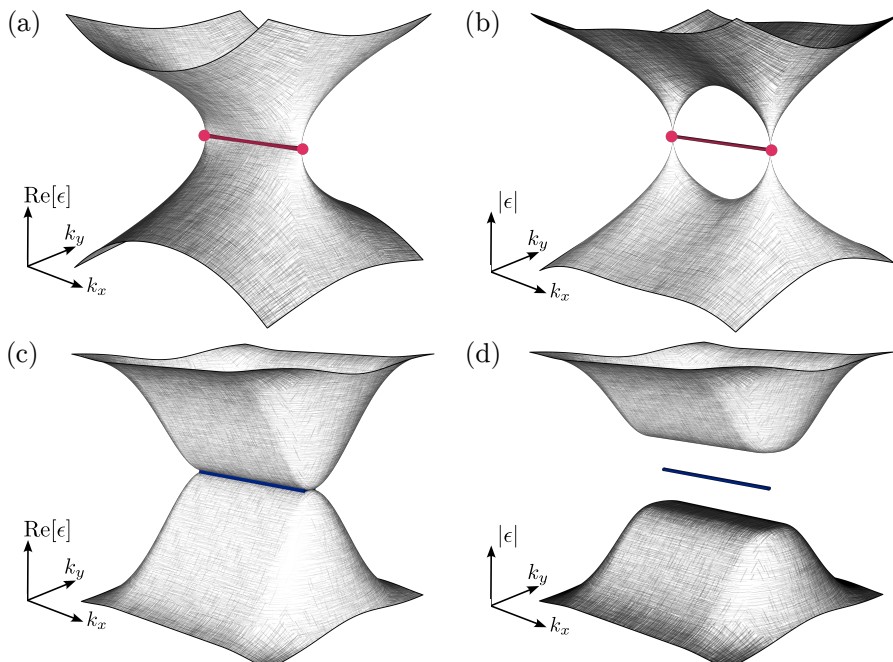

Figure 1: Illustration showing the real part and absolute value of the Riemann surface structure of non-Hermitian two-band spectra: (a), (b) show an EP2 pair connected by a branch cut identified with the Fermi arc in red. This line is referred to as a Fermi cut; (c), (d) show an open Fermi arc due to band touching in blue. In two dimensions the Fermi cut is protected by the presence of the stable EP2 pair, while the Fermi arc may disappear under perturbations.

## 1.2 Toric code ground state

A prominent application of topology is the protection of the ground states in Kitaev's toric code [21]. These states implement two logical qubits within a square lattice that is defined on a toroidal surface and has physical qubit degrees of freedom on its links. The key idea of this encoding is the definition of a Hamiltonian in terms of overlapping but commuting local operators. Due to the overlap, such a model and its states are highly non-trivial. The states can be fully determined nevertheless because all local operators commute. For the conventional toric code, this can be achieved by defining the star and plaquette operators,

$$A_s = \prod_{i \in s} \sigma_i^x \quad \text{and} \quad B_p = \prod_{i \in p} \sigma_i^z, \tag{1}$$

where the $\sigma_i^\alpha$ with $\alpha \in \{x, y, z\}$ are Pauli operators acting on the physical qubits. Here products run over the links coming together at a lattice site $s$, or over the boundary links of a plaquette $p$; cf. Fig. 2. In these terms, the toric code is governed by the Hamiltonian

$$H_{\text{tc}} = -\sum_s A_s - \sum_p B_p, \tag{2}$$

summing over all sites and plaquettes.

Measuring a plaquette or star operator yields ±1 and does not affect the model, since $[H_{\text{tc}}, A_s] = [H_{\text{tc}}, B_p] = 0$. Thus $A_s$ and $B_p$ form a basis for error detection making them so-called stabilizers [21, 28]. A state for which all of these stabilizers are measured to be +1 minimizes the energy of $H_{\text{tc}}$ making it a ground state of the system. The ground state is not unique, which can be seen by considering the following. If the physical qubits are measured

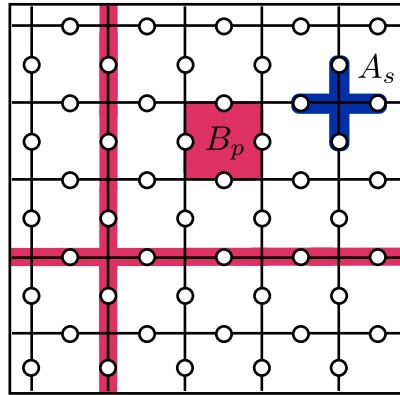

Figure 2: Illustration of the toric code on a square lattice given periodic boundary conditions. Physical qubit degrees of freedom are represented by white circles. A star and a plaquette operator, $A_s$ and $B_p$, are shown in blue and red, respectively. Non-contractible loops of flipped spins, which define the ground states, are indicated by red lines.

in the $\sigma^x$ basis, the configuration with only spin-up states achieves $A_s = +1$. So does any configuration that can be obtained from flipping spins along closed loops. Projecting these configurations onto the subspace with $B_p = +1$ results in the ground-state manifold of the toric code. Any contractible loop of flipped spins can be removed by applying plaquette operators, which leave the ground-state manifold unaffected while flipping all the physical qubits around a plaquette. Therefore the highly degenerate set of ground-state configurations is classified by four topologically distinct realizations of flipped spins along *non-contractible* loops on the toroidal surface, see the top row in Fig. 3. As such, they encode a logical two-qubit system that is protected by the topology of the non-contractible loops. The presence or absence of each possible non-contractible loop of flipped spins defines a single logical qubit.

## 2 Gapped Riemann surface topology

In the following we develop an analogy between topologically stable Fermi cuts in the spectra of non-Hermitian models and the non-contractible loops of the toric code ground states. We begin by distinguishing non-contractible lines formed by Fermi arcs from Fermi cut lines, before detailing the topological protection associated with Fermi cuts. After discussing the analogy to the ground states of the toric code, we describe how the presence of EPs can be regarded as excitations in this analogy.

### 2.1 Snapping Fermi arcs

In a preparatory first step, we identify the non-contractible loops in the toric code with Fermi *arcs*, which are closed across the Brillouin zone boundary, instead of identifying them with Fermi cuts. We stress that the presence of Fermi arcs does not change the topology of the spectral Riemann sheets. This approach illustrates that while one can find classical analogs of the toric code ground states utilizing Fermi arcs, they lack the desired topological protection. To illustrate this absence of protection, let us consider the Bloch Hamiltonian

$$H_{\text{FA}}(\boldsymbol{k}) = \begin{pmatrix} \delta + i & (1 - \cos k_x) + \frac{i}{2} \\ (1 - \cos k_x) + \frac{i}{2} & -\delta - i \end{pmatrix}, \tag{3}$$

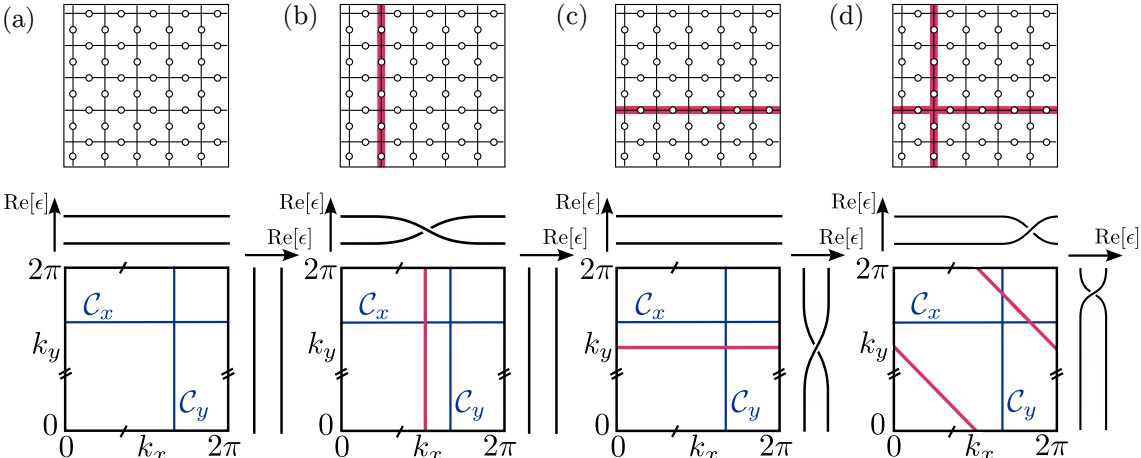

Figure 3: In the top row, the four toric code ground states are shown on the spin lattice, where flipped spins are highlighted in red. In the bottom row, we show the corresponding Fermi-cut structure in the Brillouin zone, where Fermi cuts are represented by red lines. Two distinct non-contractible loops, $\mathcal{C}_x$ and $\mathcal{C}_y$, around the different holes of the torus are indicated in blue. Along these loops the eigenenergy braids, shown next to the Brillouin zone, can be measured to determine the topological invariants.

which realizes a non-contractible Fermi arc along the $k_y$-direction of the Brillouin zone for $\delta = 0$. Any non-vanishing $\delta \in \mathbb{R}^+$ does not continuously deform the Fermi arc, but abruptly snaps it. Despite the presence of a gapped spectrum, the Fermi arc is not protected, because it can be removed without closing the gap. This is a consequence of the Fermi arc being a spectral observable only. The Riemann surface structure of the energies remains separated into distinct sheets even in the presence of a Fermi arc, compare also Fig. 1(b). Such a band touching is not topologically protected, and therefore already destabilized by small perturbations.

## 2.2 Topologically protected Fermi cuts

To construct a perturbatively stable analog of the non-contractible loops of the toric code ground states within non-Hermitian spectra, we utilize the topological protection of Fermi cuts. For the creation of a Fermi cut, a pair of EP2s must be generated and pulled apart, requiring a closing of the complex energy gap. In a non-periodic parameter space closed branch cuts cannot exist, thus a Fermi cut always depreciates to a Fermi arc when it is closed through the merger of the EP2s. This applies to the *local* creation and merger of EP2s in the periodic two-dimensional Brillouin zone as well. However, if one of the EPs is threaded through the full Brillouin zone and *across* the periodic boundary, the Fermi cut survives even after the EPs are merged again. This results in a fully non-degenerate system with a finite gap, which prevents the perturbative creation of EP pairs, thus protecting the Fermi cut. The resulting closed branch cut runs through the whole Brillouin zone along a non-contractible loop. The threading of the EP changes the structure of the spectral Riemann surface non-trivially, resulting in a fully non-degenerate continuously-connected energy structure. We stress that it is the toroidal topology of the Brillouin zone that facilitates such a closed Fermi cut.

The topological protection of the Fermi cuts after threading and merging the EPs can be understood from the perspective of eigenenergy braids. These braids are defined on closed lines around the holes of the toroidal Brillouin zone, cf. the bottom row in Fig. 3. A Fermi cut results in a crossing of the braids orthogonal to the cut. Perturbations cannot change the non-trivial braids of single Fermi cuts, ensuring their stability [29]. This is similar to the

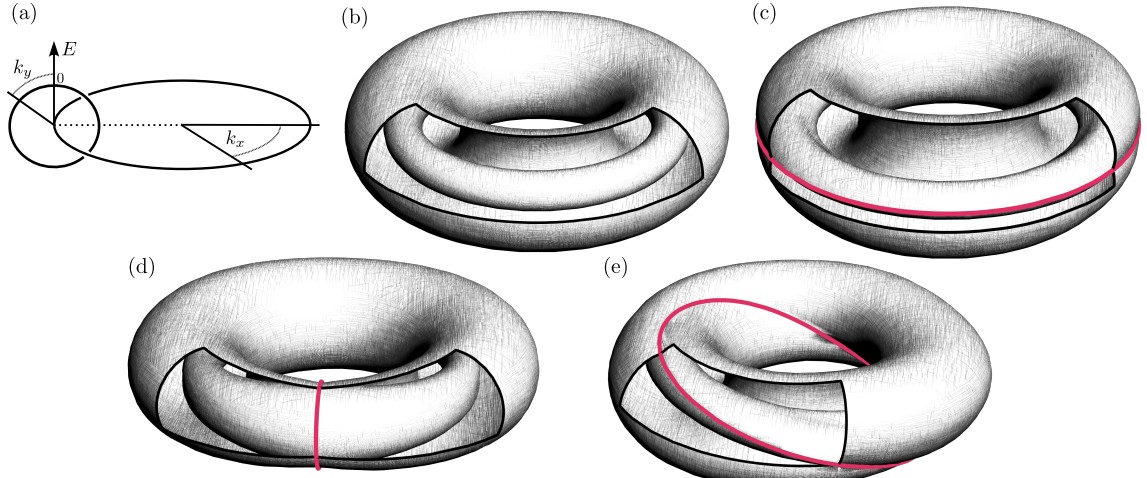

Figure 4: The real part of the spectral Riemann surface structures over the Brillouin zone is shown for the topologically distinct configurations. (a) Coordinate system of the toroidal Brillouin zone, where the two quasi-momenta $k_x, k_y$ correspond to the two angles in toroidal coordinates and the radius encodes the real part $E$ of the complex-valued spectrum; (b)-(e): Topologically distinct fully non-degenerate configurations of the non-Hermitian model $H_{\text{FC}}(\boldsymbol{k})$ analogous to the ground states of the toric code. The states are distinguished by the topological invariants $(m_x, m_y)$: (b) $(0,0)$, (c) $(0,1)$, (d) $(1,0)$, (e) $(1,1)$.

braid protection of third-order EPs observed in Ref. [30]. In contrast to previous works, which consider braids along loops that encircle the full Brillouin zone, the topological protection here arises from braids along non-contractible loops. Unlike Ref. [30], where only isolated higher-order EPs result in non-trivial braids, this allows to capture non-contractible Fermi cuts.

The presence or absence of non-trivially closed Fermi cuts results in a different connection of the Riemann energy sheets. This connectedness describes the (im-) possibility to change between sheets by following a non-contractible loop $\mathcal{C}$ in the Brillouin zone. If the eigenenergy crosses an even number of Fermi cuts, it returns to its initial value after traversing the parametric loop $\mathcal{C}$–the sheets are disconnected along this direction. When crossing an odd number of Fermi cuts, the sheets are connected instead and the eigenenergy does not return to the initial value. By distinguishing whether the Riemann sheet is connected along both directions, we define four topologically distinct configurations of the system. These configurations are only well-defined for a gapped non-degenerate system without EPs present. Tuning the system from one of these states to another thus requires the merger of all EPs after the threading procedure. To distinguish the configurations we define two topological invariants, which are derived from the permutations of the eigenenergies along non-contractible loops through the Brillouin zone. The invariant $m_\alpha = 0$ is associated with non-connected and $m_\alpha = 1$ with connected Riemann sheet structures, where $\alpha \in \{x, y\}$ indicates whether the path $\mathcal{C}$ runs along the $k_x$ or $k_y$ direction. Specifically, a non-vanishing invariant $m_\alpha$ corresponds to a non-trivial permutation $\pi_\alpha$ of the two energies, that is, exchanging the eigenvalues $\epsilon_1$ and $\epsilon_2$ of the two-sheeted Riemann surface:

$$m_\alpha = 1 - \delta_{\pi_\alpha(1),1}, \tag{4}$$

in terms of the Kronecker delta $\delta_{i,j}$. For example, if a non-contractible loop in $k_\alpha = k_x$ direction crosses an odd number of Fermi cuts, the eigenenergies exchange and we find $\pi_x(1) = 2$, and thus $m_x = 1$. This establishes the $\mathbb{Z}_2 \times \mathbb{Z}_2$ topological invariant for the gapped non-Hermitian spectra.

Based on the four configurations distinguished by the combinations of topologically pro­tected closed Fermi cuts, we construct the analogy to the ground-state manifold of the toric code in an exemplary model. Any closed Fermi cut is mapped to a non-contractible loop of flipped spins in the toric code, and the four different ground states are represented by $(m_x, m_y) \in \{(0,0),(1,0),(0,1),(1,1)\}$. The ground-state manifold is realized by the tunable non-Hermitian system with the Bloch Hamiltonian

$$H_{\text{FC}}^{(m_x, m_y)}(\boldsymbol{k}) = \begin{pmatrix} 3 - \cos s_{\boldsymbol{k}} - \cos a_{\boldsymbol{k}} & \sin s_{\boldsymbol{k}} - 3i(1 - \cos s_{\boldsymbol{k}}) \\ \sin s_{\boldsymbol{k}} - 3i(1 - \cos s_{\boldsymbol{k}}) & -3 + \cos s_{\boldsymbol{k}} + \cos a_{\boldsymbol{k}} \end{pmatrix}, \tag{5}$$

where $s_{\boldsymbol{k}} = m_x k_x + m_y k_y$ and $a_{\boldsymbol{k}} = m_y k_x - m_x k_y$. Figure 4 showcases the four topologi­cally distinct configurations of the non-Hermitian toric-code analog and highlights the non­trivially closed Fermi cuts, which run along the curves defined by $s_{\boldsymbol{k}} = (\pi \mod 2\pi)$ for the given system. When tuning continuously between two topologically distinct configurations, $H(\lambda) = (1 - \lambda) H_{\text{FC}}^{(m_x, m_y)} + \lambda H_{\text{FC}}^{(m'_x, m'_y)}$ with $\lambda \in [0,1]$ and $(m_x, m_y) \neq (m'_x, m'_y)$, the complex gap closes at a value $\lambda_-$ and opens only after a finite interval at $\lambda_+ > \lambda_-$. Within the inter­val $(\lambda_-, \lambda_+)$ a pair of EP2s is threaded through the Brillouin zone, changing the topological structure of the Riemann sheets.

## 2.3 Exceptional points as excitations

The topologically distinct configurations of the non-Hermitian model are only defined for non­degenerate systems in which no EPs are present. In the following, we show that EPs can be regarded as excitations, analogous to the role of defective sites and plaquettes in the toric code.

The excitations in the conventional toric code are sites $s$ or plaquettes $p$, whose respective stabilizers, $A_s$ or $B_p$, are measured to be $-1$ [21]. This raises the energy of the system above the lowest possible energy state in which all stabilizers are $+1$. By flipping a single physical qubit in the $\sigma^x$-basis, a pair of star defects called electric charges $e$ is created on the adjacent sites. Similarly, flipping in the $\sigma^z$-basis creates two plaquette defects, called magnetic charges $m$, on the plaquettes that share the edge. These excitation pairs can be pulled apart and moved around the lattice. They stay connected by a so-called flux line, which is an open line of flipped spins. Excitations of the same type annihilate when brought together, and the flux lines become closed loops, so that the system returns to a ground state; nontrivial flux threading tunes the system between different states in the ground-state manifold. Alike charges exchange bosonically, while different charges commute mutually anyonic. Bringing two different charges together results in a dyon, which is a fermionic composite excitation.

In the non-Hermitian toric-code analog the topologically distinct configurations are char­acterized by a finite gap. Transitioning between these configurations requires the spectral gap to close in the form of EPs. Similar to excitations in the conventional toric code, they appear pairwise and are connected by an open Fermi cut, which plays the role of a flux line. In this sense, EPs can be interpreted as excitations of the non-Hermitian system. For a two-band model, the only possible types of excitation are therefore EP2s. Moreover, the EP2s carry a topological charge $\nu$, given by the spectral winding number

$$\nu = -\oint_{\mathcal{C}} \frac{d\boldsymbol{k}}{2\pi} \cdot \nabla_{\boldsymbol{k}} \arg[\Delta \epsilon(\boldsymbol{k})] = \pm \frac{1}{2}, \tag{6}$$

where $\mathcal{C}$ is a closed path encircling the EP2 and $\Delta \epsilon(\boldsymbol{k})$ is the difference between the energy sheets that coalesce at the EP2 [19, 20]. As a result, excitations in the non-Hermitian system carry additional structure compared to the conventional toric code, because only EP2s of op­posite topological charge can annihilate. Combining EP2s of identical charge, on the other hand, results in an unstable EP2 composite with additively combined topological charge.

In contrast to the conventional toric code, a non-Hermitian two-band model, while realizing four distinct non-degenerate configurations, allows for only one instead of three excitation types. The implementation of multiple distinct excitations requires systems with additional bands in the Bloch Hamiltonian. Adding a third band, for instance, results in three types of EP2s, which are the different possible branch points between pairs of eigenenergy sheets. When bringing two different EP2s together, EP3s emerge as composite excitations [31, 32]. In general, extending this to $n$-band systems allows for $\sum_{j=2}^{n} \binom{n}{j} = 2^n - (n+1)$ different types of excitations. These are again associated with topological charges, determined by the winding of the eigenenergies around them [33, 34].

At the same time, added bands affect the possible topologically distinct configurations of the non-Hermitian multi-band model, which comprise any fully non-degenerate spectral structure. The number of topologically distinguishable Riemann surface structures increases, because additional non-contractible Fermi cuts along the $k_x$ and $k_y$ direction are possible between any two bands. However, intersecting Fermi cuts induce EP3 degeneracies if a single energy sheet is part of both cuts. Therefore not all combinations of Fermi cuts result in gapped non-Hermitian spectra. A set of topological invariants can be defined for each non-degenerate Riemann sheet structure by generalizing Eq. (4) to all pairs $(p, q)$ of energy sheets and repeated application of the permutation $\pi_\alpha$:

$$m_\alpha^{pq} = 1 - \delta\Big[ \sum_{j=1}^{n-1} \delta\big[ p - \pi_\alpha^j(q) \big] \Big],\qquad(7)$$

where $\delta[n]$ denotes the Kronecker delta $\delta_{0,n}$. This establishes the topologically distinct non-degenerate spectral surfaces as a subspace of the space with a $(\mathbb{Z}_2 \times \mathbb{Z}_2)^m$ topological invariant, where $m = \sum_{j=1}^{n-1} j = \frac{1}{2}(n-1)n$.

Overall, extending the non-Hermitian model to multi-band systems goes beyond the initial analogy to the toric code and allows for the realization of multiple distinct excitations.

# 3 Implementation on metasurfaces and in single-photon interferometry

The implementation of different topologically protected configurations requires a high degree of parametric tunability over a toroidal parameter space. Measuring the connectedness of the Riemann surface structure on the other hand is achievable, for example by sampling the real part of the spectrum along non-contractible loops in the Brillouin zone. We remark that conclusions about the connectedness based on state evolution are not possible, since the adiabatic theorem does not hold for the non-Hermitian models discussed here [35–37]. Prominent platforms providing the necessary tunability are optical or plasmonic metasurfaces, which rely on artificial units cells defined in periodic arrays of nanoantennas [38, 39]. Loss generates non-Hermitian contributions to the dynamics of these systems. Such setups are capable of realizing the two-band model $H_{\text{FC}}(\boldsymbol{k})$ in a two-orbital square lattice given full control over the onsite terms and hoppings up to next-nearest-neighbor distance. Mechanical metamaterials, in which individual oscillators are driven depending on the state of the system, or single-photon interferometry, recently established to study non-Hermitian topological features [40], provide other feasible platforms for the observation of EP pair creation and their threading through the Brillouin zone. They are thus capable of resolving the transition between topologically distinct states.

# 4 Conclusion

We have demonstrated topological structures tied to the presence an energy gap in the spectral Riemann surfaces of non-Hermitian systems. Pairs of EPs emerge as uniquely non-Hermitian features in these complex-valued energy structures. We introduce Fermi cuts as the branch cuts connecting EPs along lines with degenerate real energy parts. These cuts can be closed non-trivially by separating the EP pair, and recombining it across the Brillouin-zone boundary. The EPs then annihilate, leaving a non-contractible closed Fermi cut. Building on an analogy to Kitaev's toric code, these Fermi cuts resemble non-contractible closed defect lines on the toroidal Brillouin zone of a two-dimensional lattice. Four topologically distinct configurations of the non-Hermitian system are characterized by a fully non-degenerate spectrum and EPs are interpreted as excitations. Non-Hermitian multi-band systems facilitate multiple different excitations and the presence or absence of the closed Fermi cuts establishes topologically distinct configurations. The energy spectrum behaves as a classical object, so that this analogy does not inherit the full quantum-mechanical properties of the toric code. It instead gives rise to Riemann sheet structures beyond the conventional toric-code dimension. The resulting configurations are topologically protected and remain stable under perturbation, due to finite gaps in the complex energy spectra. This framework establishes a novel approach to utilize the topological features inherent in non-Hermitian systems.

## Acknowledgements

We are grateful to Joost Slingerland for his lectures during the Young Researchers School 2024 in Maynooth and to J. Lukas K. König for fruitful discussion.

**Funding information**      A.M., A.F. and F.K.K acknowledge funding from the Max Planck Society's Lise Meitner Excellence Program 2.0.

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
