# Peer review of "Spectral Riemann Surface Topology of Gapped Non-Hermitian Systems"

_SciPost Physics_

## Round 1 · Referee Report · Anonymous (Referee 1) · 2025-9-21

Strengths

1 - Interesting idea of the connection between the non-Hermitian band topology and the toric code

Weaknesses

1 - The manuscript is extremely hard to follow for the general theoretical-physics audience

2 - The discussion is too abstract in the sense that it is mainly presented in words, using extremely long paragraphs, instead of explicit mathematical expressions

3 - Illustrations are not accessible without the corresponding formulas

Report

The manuscript presents a nice idea of "mapping" the topological features of the toric code to a general structure of bands in non-Hermitian Bloch Hamiltonians. This is a very nice parallel proposed by the authors, which could be implemented in future studies in the highly topical field of topology in non-Hermitian physics. However, the presentation in the manuscript is designed to target only a rather narrow community (like the authors of Ref. [30]). Unfortunately, the authors of the present manuscript did not invest efforts into making the article self-contained. The manuscript is a flow of text (with very long paragraphs), operating with specific terms, most of which are not properly defined or explained in this text. Although references are given for most of the notions appearing in the manuscript, random readers will not be able to complete reading the article if they seriously consult all those references. The figures, which are intended to illustrate the notions and objects discussed in the manuscript, do not serve this purpose; on the contrary, they enhance the confusion. Already the very first paragraph introduces the notion of "energy Riemann surfaces." Of course, the general theoretical audience should know what Riemann surfaces are; however, when talking about energy Riemann surfaces for non-Hermitian systems, the authors do not give a definition of this primary object, sending the reader to Refs. [4,18,24-27]. Explaining the relation between the Fermi arcs and Fermi cuts, the authors try to illustrate this in Fig. 1, which does not explain anything. For example, why do the regions where Re[epsilon] exists not exist in the |epsilon| plot? The authors should give explicit mathematical expressions for Re[epsilon(kx,ky)] and Im[epsilon(kx,ky)] for some simple illustrative non-Hermitian models, and then, referring to the mathematical properties of these explicit formulas for the band structure, explain the notions they discuss in the manuscript. This should also be done for the Hamiltonians (3), (5) addressed later in the manuscript: the mathematical structures should be introduced and analyzed step-by-step. Without this, it is totally unclear why Fig. 4 has to do with Eq. (5), for example. Figure 4 appears as an illustration in a popular-science presentation showing that the mug with a handle is topologically equivalent to a donut. The only part of the manuscript that contains sufficient information is the description of the toric code. All other parts must be made much more explicit: this is a scientific paper and not a poetry book.

Requested changes

1 - add explicit mathematical expressions for the band structure in the examples considered

2 - Structure the narrative along the sequence of the mathematical expressions

3 - Add more figures illustrating the particular steps

Recommendation

Ask for major revision

---

## Round 1 · Referee Report · Anonymous (Referee 2) · 2025-11-2

Strengths

1- The manuscript presents an interesting analogy between cuts in the Brillouin zone of a non-Hermitian model and the toric code 2 - Their work motivates new links between single particle non-Hermitian Hamiltonians and topological many-body models

Weaknesses

1 - The manuscript is hard to follow for non-experts in its current form
2 - The manuscript would benefit from presenting the idea in a more intuitive picture at least at the beginning
3 - The implementation section is quite short and high level, describing generic platforms to emulate non-Hermitian phenomena. It could be beneficial that the authors provide a concrete realistic calculation for it

Report

The authors present a connection between the spectra of non-Hermitian systems and the spectra of the toric code. This mapping is related with different cuts in momentum space of the non-Hermitian spectra, showing exceptional points. While the results may be of interests to expert, I believe that the manuscript would strongly benefit from rewriting it in a more accessible way, including phrasing out more clearly the relevance of these results for non-Hermitian physics. As the manuscript currently is, the results are hard to follow, even for readers with background on non-Hermitian phenomena. I write some concrete suggestions below.

Requested changes

1- I would encourage the authors to expand some paragraphs in smaller sections, more clearly outlining the important steps in the calculation 2- If a similar analogy could be done with a 1D non-Hermitian model, I would encourage the authors to include it as it can be greatly helpful to understand their idea. The physics of that case would be of course different from the toric code, yet I believe that it would substantially clarify the essence of their manuscript. 3- The authors start explaining the toric code, and later moving towards their non-Hermitian case. From my perspective it would be more clear if they make a complete explanation of their non-Hermitian case, and then later on they show the analogy with the toric code 4- Section 2.3 would benefit from a concrete example, probably accompanied with a figure. As it stands, the explanation about the excitations is hard to follow.

Recommendation

Ask for major revision

---

## Editorial Decision

resubmitted